

# Photodegradation and biodegradation of dissolved organic matter on the surface of the Greenland Ice Sheet

Miranda J. Nicholes[1], Christopher Williamson[1], Martyn Tranter[1], Alexandra Holland[1], Marian Yallop[2], Alexandre Anesio[3]

[1]Bristol Glaciology Centre, School of Geographical Sciences, University of Bristol, Bristol, BS8 1SS, UK
[2]School of Biological Sciences, University of Bristol, Bristol, BS8 1TQ, UK
[3]Department of Environmental Science, Aarhus University, Roskilde, 4000, Denmark

*Correspondence to*: Miranda Nicholes (miranda.nicholes@bristol.ac.uk)

## Abstract

The surface (supraglacial) environment of the Greenland Ice Sheet (GrIS) is an active site for the storage, transformation and transport of carbon, which is driven by extremely high levels of solar radiation throughout the ablation season. Within the south west of the GrIS, blooms of Streptophyte micro-algae (hereafter 'glacier algae') at abundances of $\sim 10^5$ cell mL$^{-1}$ dominate primary production in the surface ice and provide dissolved organic matter (DOM) to the heterotrophic bacterial community. Glacier algae contain photoprotective secondary phenolic pigment that comprises a large proportion of the cell ($\sim 4\%$ of the dry weight) and could represent a substantial, additional carbon source for the heterotrophic community. The transformation and degradation of DOM by solar radiation (photodegradation) and heterotrophic communities (biodegradation) represent two crucial controls on DOM composition and quantity; however, the influence of these processes within the surface ice is yet to be constrained. This study therefore assessed responses in the composition and quantity of two carbon sources (glacier algae secondary pigment and surface ice DOM) following exposure to UV, PAR, UV+PAR (photodegradation) and subsequent incubation with bacterial communities isolated from the ambient environment (biodegradation). Our results indicate that exposure to predominantly UV radiation altered the composition of glacier algal pigment and surface ice DOM; however, the quantity of DOM remained constant. Biodegradation caused the greatest changes to both DOM composition and quantity, particularly in surface ice DOM. Secondary pigment extracted from glacier algae was not a highly bioavailable source of carbon and did not support significant growth of surface ice heterotrophic bacterial communities. Conversely, low molecular weight compounds in surface ice DOM were rapidly utilised by heterotrophic bacteria supporting between a 3 and 9-fold increase in bacterial abundance over a 30-day incubation. We found that photodegradation of glacier algal pigment and surface ice DOM did not influence heterotrophic consumption. Photodegradation and biodegradation of DOM in the surface ice habitat are likely intimately linked and act as fundamental controls on the composition and quantity of DOM exported to downstream environments.

## 1 Introduction

Dissolved organic matter (DOM) is ubiquitous across all aquatic systems and comprises a chemically diverse range of compounds (Baker and Spencer, 2004; Coble et al., 2014; Kellerman et al., 2018). DOM has multiple





functions within ecosystems including the provision of substrates for heterotrophic organisms; mobilising organic
and inorganic pollutants; and influencing the bioavailability of nutrients (Aiken, 2014; Baker and Spencer, 2004).
DOM originates from either autochthonous (i.e. in situ production) or allochthonous (i.e. terrestrial) sources with
varying levels of reactivity and composition. For example, autochthonous DOM typically comprises of low
molecular weight (LMW) compounds such as amino acids and carbohydrates, which are highly bioavailable
(Amado et al., 2015; Murphy et al., 2008; Stedmon and Markager, 2005). In contrast, allochthonous carbon
usually comprise of high molecular weight (HMW), aromatic compounds of a more recalcitrant nature (Amado
et al., 2015; Murphy et al., 2008). The composition and quantity of DOM fundamentally controls the flow of
carbon and energy through aquatic ecosystems, shaping the structure of the food web.

The composition and quantity of DOM in aquatic ecosystems is primarily controlled by two processes:
photodegradation and biodegradation (Benner and Kaiser, 2011; Fasching et al., 2014; Hansen et al., 2016).
Chromophoric DOM (CDOM) comprises a proportion of total DOM that is highly reactive on exposure to solar
radiation (Coble, 2007). Consequently, high energy ultra-violet (UV) radiation can strongly influence carbon
cycling by degrading CDOM into a range of species including dissolved inorganic carbon, smaller organic carbon
compounds and reactive oxygen species (ROS; e.g. hydroxyl radicals) (Coble, 2007; Lindell et al., 1995; Stefan
et al., 2000; Tranvik and Bertilsson, 2001). DOM is a vast resource of biologically available organic carbon that
is consumed and transformed by heterotrophic bacteria (biodegradation) (Fellman et al., 2010). Carbon is utilised
by heterotrophic bacteria through two different processes: i) complete mineralisation of carbon to obtain energy
i.e. respiration; ii) incorporation into microbial biomass (Ducklow, 2000). Photodegradation of DOM can strongly
influence the interaction between heterotrophic communities and carbon within ecosystems exposed to high levels
of solar radiation (Anesio and Granéli, 2004; Stefan et al., 2000; Tranvik and Bertilsson, 2001; Tranvik and
Kokalj, 1998). The photodegradation of complex, recalcitrant DOM into more bioavailable compounds has been
found to stimulate bacterial production (Amado et al., 2015; Lindell et al., 1995; Zepp and Moran, 1997); however,
radiation can also remove LMW, bioavailable compounds from the DOM pool (Tranvik and Kokalj, 1998).
Additionally, the generation of ROS via photodegradation damages bacteria, reducing production (Holzinger and
Lütz, 2006; Ravanat et al., 2001). It is thought that the relative influence of photodegradation on bacterial
production is determined by the balance of these processes and relates to the source of DOM; i.e. photodegradation
tends to reduces bioavailability of algal-derived DOM but increases bioavailability of allochthonous DOM
(Amado et al., 2015). Thus, in ecosystems exposed to high levels of solar radiation, DOM composition and
quantity is controlled by two highly interlinked processes.

The surface ice of the Greenland Ice Sheet (GrIS) hosts abundant and active algal and bacterial communities
which influence both regional and global carbon cycling (Nicholes et al., 2019; Perini et al., 2019; Williamson et
al., 2018; Yallop et al., 2012). During the ablation season, the surface ice is exposed to extreme levels of
photosynthetically active radiation (PAR; ~ 1700 μmol photons $m^{-2}$ $s^{-1}$ on a cloudless day) and UV radiation.
Solar radiation drives high levels of carbon fixation (0.35 – 1.12 mg C $L^{-1}$ $d^{-1}$) predominantly by blooms of highly
pigmented glacier algae (Musilova et al., 2017; Williamson et al., 2018; Yallop et al., 2012) that are particularly
prevalent in the so-called 'Dark Zone' in the south west of the ice sheet. Glacier algae are well-adapted to the
extreme light conditions and synthesise an abundant secondary phenolic pigment (purpurogallin carboxylic acid-





6-*O*-β-ᴅ-glucopyranoside) that exhibits maximal absorbance in UV-B wavelengths and thus protects the cells
from photo-damage (Remias et al., 2012; Williamson et al., 2020). This pigment comprises a relatively high
proportion of glacier algal cellular content (~ 4 % of the dry weight; Williamson et al. 2020) and likely reaches
high concentrations during bloom events when algal abundance peaks at $10^4$ cells $mL^{-1}$ (Williamson et al., 2018;
Yallop et al., 2012). Glacier algal pigment, comprised predominantly of phenolic based compounds (Remias et
al., 2012), could therefore represent a vast source of organic carbon within the GrIS supraglacial environment if
released during cell lysis due to natural senescence or viral infection, providing an important energy source for
the diverse and active bacterial communities that numerically dominate surface ice (Hodson et al., 2008; Nicholes
et al., 2019; Stibal et al., 2015). These communities are thought to utilise predominantly algal-derived DOM to
achieve rates of bacterial production (BP) ranging 0.03 – 0.6 µg C $L^{-1}$ $h^{-1}$ (Nicholes et al., 2019; Yallop et al.,
2012); approximately 30-times less than primary production (Nicholes et al., 2019; Yallop et al., 2012).

Despite the presence of active autotrophic and heterotrophic communities, there remains several critical
knowledge gaps regarding carbon cycling within the high-light supraglacial surface ice environment. These
include the potential impact of photodegradation on surface ice DOM and subsequent heterotrophic utilisation;
the susceptibility of glacier algal secondary phenolic pigment to photodegradation (given its maximum absorbance
in UV-B wavelengths) and its bioavailability to heterotrophic bacteria; and the components of surface ice DOM
that are degraded by heterotrophic bacteria. Thus far, investigations into DOM cycling in glacial environments
have revealed that photodegradation produces a slight increase in bioavailable DOM in Antarctic snow samples
(Antony et al., 2018). However, DOM biodegradation was found to produce photoreactive compounds
highlighting the interlinked nature of these two processes (Antony et al., 2017, 2018). Here, we hypothesise that
photodegradation may alter the composition and bioavailability of organic carbon to heterotrophic bacterial
communities, subsequently impacting biodegradation pathways in surface ice. This investigation therefore aimed
to constrain changes in the composition and quantity of both ambient surface ice DOM and glacier algae secondary
phenolic pigment (representing a potentially abundant and refractory DOM source) following exposure to UV
and PAR radiation, and their subsequent bioavailability to surface ice bacterial communities.

## 2    Methodology

An incubation experiment was conducted to determine the degree to which glacier algal phenolic pigment
(hereafter 'pigment') and surface ice dissolved organic matter (DOM) degrade when exposed to combinations of
ultraviolet (UV) and photosynthetically active radiation (PAR) radiation. The bioavailability of pigment and
surface ice DOM following photodegradation was subsequently assessed via incubations with a heterotrophic
bacterial community isolated from surface ice samples.

### 2.1    Surface ice preparation

Surface ice was obtained during the 2017 Black and Bloom field campaign (31st May to 1st July 2017) from the
primary ice camp established within the GrIS ablation zone approximately 35 km from the south-western ice sheet
margin (67 ° 04'43.3" N, 49 ° 20'29.7" W), adjacent to the S6 PROMICE weather station. The top 2cm of surface





ice containing a high algal coverage (~ $10^4$ cells mL$^{-1}$) (Williamson et al., 2018) was sampled using a clean ice
saw and transferred into 5 WhirlPak bags. Each WhirlPak bag contained approximately 1.5L of frozen surface ice
and was transported back to the University of Bristol and stored at -20 °C. Prior to photodegradation, surface ice
was thawed over 24 hours at 3 °C and filtered through a 0.2 μm polyethersulfone (PES) filter (pre-flushed with
50 mL Milli-Q) to remove bacteria, fungi and glacier algae. The filtrate was homogenised and decanted into 250
mL pre-combusted Pyrex crystallising basins (n = 20).

## 2.2  Glacier algal pigment preparation

Secondary phenolic pigment was extracted from glacier algal cells present in surface ice sampled from the primary
ice camp as outlined previously. Between 100 - 200 mL of sample (n = 45) was filtered onto combusted GF/F
filters, frozen and transported to the University of Bristol. Filters were freeze-dried for 24 hours and water-soluble
pigments extracted in 5 mL of Milli-Q water, following Remias *et al.*, (2012) and Williamson *et al.*, (2018). A
phase separation with n-hexane was performed to remove non-polar constituents from the raw extract, which was
subsequently stored frozen at -20 °C until being defrosted over 12 hours prior to use. Pigment was added to Milli-
Q water to a 1:100 v/v dilution, homogenised and divided into 250 mL pre-combusted Pyrex crystallising basins
(n = 20).

## 2.3  Photodegradation

To stimulate photodegradation in the pigment and surface ice, both carbon sources were exposed to combinations
of UV and PAR over a 48-hour period. Short-wavelength, high-energy UV-radiation is widely reported as
responsible for inducing photodegradation in aquatic ecosystems (Amado et al., 2015; Bertilsson and Tranvik,
1998; Stefan et al., 2000; Tranvik and Bertilsson, 2001). Ultra-violet bulbs (25W, 220-240V, Exo Terra, Canada)
and LED bulbs (Prolite, UK) were used to create 3 different light treatments: UV, PAR (432 μmol photons m$^{-2}$ s$^{-1}$
) or UV plus PAR (UV+PAR). Pigment and surface ice were exposed to UV, PAR and UV+PAR (n = 5 per light
treatment per carbon source) at 3 °C for 48 hours. Light exposure for 48 hours produced 840 kJ m$^{-2}$ of UV-A, 412
kJ m$^{-2}$ of UV-B and 16,456 kJ m$^{-2}$ of PAR, equivalent to approximately 2, 10 and 4 days of exposure on the
surface of the GrIS respectively (https://pvlighthouse.com.au/). To serve as a control, pigment and surface ice
samples were wrapped in foil and kept in total darkness during the exposure period ("DARK", n = 5). In addition,
a Milli-Q control (n = 5 per light treatment) was incubated alongside pigment and surface ice to detect any
contamination. Following photodegradation, dissolved organic matter composition (UV-Vis and fluorescence
spectroscopy) and quantity (dissolved organic carbon concentration) was assessed.

## 2.4  Biodegradation

To determine the bioavailability of photodegraded pigment and surface ice, carbon sources were inoculated with
bacteria and incubated for 31 days. Bacterial cultures were established from surface ice from the primary ice
camp, sampled as outlined previously. The surface ice was thawed at 3 °C over 48 hours, decanted into a pre-
combusted 1000 mL beaker, covered with furnaced foil and sonicated for 2 minutes to facilitate cell detachment





from particles (Bradley et al., 2016). To isolate bacteria, surface ice was filtered through a combusted GF/D filter
(Whatman, USA) and stored at 3 °C. Bacteria were inoculated to pigment, surface ice and Milli-Q control at a 10
% v/v final concentration in pre-combusted 30 mL amber glass vials, maintaining a headspace. Bacterial
abundance at the start of the incubation averaged $3.7 \pm 0.2 \times 10^4$ cells mL$^{-1}$ and did not differ significantly across
pigment, surface ice or Milli-Q control samples.

To control for and examine the potential influence of nutrient limitation on carbon consumption and modification
during incubations, half of all replicates received additions of inorganic nitrogen ($NH_4NO_3$) and phosphorus
($KH_2PO_4$) across all treatments at final concentrations of 30 µM L$^{-1}$ and 10 µM L$^{-1}$, respectively; representing 10-
times ambient concentrations reported from the surface of the GrIS (< 1 µM P L$^{-1}$ Hawkings *et al.*, 2016; 1.3 µM
DIN L$^{-1}$ Wadham *et al.*, 2016).  Incubations proceeded at 3 °C in the dark for a period of 31 days, with destructive
sampling at days 0, 3, 6, 10, 17 and 31 to assess biodegradation impacts to DOM composition and quantity within
incubations, relative to bacterial abundance and biovolume.

**2.5    UV-Vis spectroscopy**
Analysis of CDOM via spectroscopy can provide information regarding DOM aromaticity, sources and reactivity
(Li and Hur, 2017). Accordingly, UV-Vis spectroscopy of the pigment and surface ice was conducted following
both photodegradation and biodegradation steps of our experiment. Absorbance spectra were obtained using a
Varian Cary 60 UV/Vis spectrophotometer (Agilent Technologies, USA) with scans run over wavelengths ranging
from 200 – 800 nm at 2 nm intervals. Absorption data is expressed as absorption coefficients, calculated following
Eq. (1):
$a\,(\lambda) = 2.303\,A(\lambda)/l$ (1)
where $a\,(\lambda)$ is absorption coefficient (m$^{-1}$), $A(\lambda)$ is the raw absorbance and l is the cuvette pathlength (m).
Absorbance indices utilised are summarised in Table 1. Specific ultraviolet absorbance at 280 nm (SUVA$_{280}$) is a
proxy for total aromaticity as electron transition occurs within this absorbance region for phenolic arenes, benzoic
acids, aniline derivatives, polyenes and polycyclic aromatic hydrocarbons with two or more rings (Uyguner and
Bekbolet, 2005). Although SUVA indices give an indication of the relative proportion of aromatic DOM, the
relative reactivity of these compounds cannot be  inferred (Weishaar et al., 2003). We therefore combined UV-
Vis and fluorescence spectroscopy to provide information of composition and bioavailability of DOM.

**2.5.1    Fluorescence spectroscopy**
A small fraction of CDOM, known as fluorescent DOM (FDOM), emits fluorescence energy when excited by
photons at specific energies (Li and Hur, 2017). Fluorescence spectroscopy therefore characterises FDOM
providing information on source, reactivity and composition (Aiken, 2014; Coble, 1996; Coble et al., 1990, 2014).
To minimise the effects of temperature, samples were left to reach room temperature before measurements were
undertaken. Fluorescence scans were conducted using a Varian Cary Eclipse Fluorescence Spectrophotometer
(Agilent Technologies, USA) scanning over excitation wavelengths 250 – 450 nm at 5 nm intervals and 300 –





600 nm emission wavelengths at 2 nm intervals. Excitation- Emission Matrices (EEMs) were processed using the
StaRdom package (Pucher et al., 2019) in R (R Development Core Team, 2019). EEMs were blank corrected
using EEMs from daily Milli-Q scans, corrected for inner-filter effects using absorbance scans and Raman
normalised. Fluorescence indices derived from EEMs are summarised in Table 1. The fluorescence intensity of
commonly identified peaks in natural waters, summarised in Table 2, was identified in sample EEMs.


**Table 1: Absorbance and fluorescence indices utilised in this study. LMW= low molecular weight. Adapted from**
**Hansen et al., (2016)**

| Indices | Calculation | Proxy | Reference |
|---|---|---|---|
| **Specific ultraviolet absorption (SUVA) E.g. SUVA$_{254}$, SUVA$_{280}$, SUVA$_{365}$ (L mg$^{-1}$ m$^{-1}$)** | Absorption coefficient at given wavelength in the ultraviolet region divided by DOC concentration | A higher number is generally associated with greater aromatic DOC content | (Weishaar et al., 2003) |
| **Specific visible absorption (SVA) E.g. SVA$_{440}$ (L mg$^{-1}$ m$^{-1}$)** | Absorption coefficient at 440 nm divided by DOC concentration | A higher number is generally associated with greater aromatic DOC content | (Chin et al., 1994) |
| **Absorption coefficient at 440 nm (A$_{440}$) (m$^{-1}$)** | Absorption coefficient at 440 nm | Indicates the colour and is therefore a proxy for concentration of humic acid | (Fasching et al., 2014) |
| **Spectral slope λ300-700 nm (S$_{300-700}$) (nm$^{-1}$)** | Spectral slope within log-transformed spectra between 300 and 700 nm | Generally a higher value indicates LMW and/or decreasing aromaticity | (Helms et al., 2008) |
| **Humification index (HIX)** | The area under the emission spectra 435-480 nm divided by the peak area 300-345 nm + 435 – 480 nm, at excitation wavelength 254 nm | Gives an indication of the degree of humification. Higher values indicate an increasing degree of humification | (Ohno, 2002) |
| **Fluorescence index (FI)** | The ratio of emission wavelengths at 470 nm and 520 nm, obtained at an excitation wavelength of 370 nm | Identified the relative contribution of terrestrial and microbial sources to the DOM pool. Increasing values suggests a microbial source | (McKnight et al., 2001) |






**Table 2: Summary of commonly identified fluorescence peaks of aquatic DOM adapted from Fellman, Hood and Spencer, (2010). Peaks were originally identified by Coble et al., (1990), (2014); Coble, (1996); Coble, Del Castillo and Avril, (1998); Murphy et al., (2008). HMW = high molecular weight; LMW = low molecular weight.**

| Peak name | Excitation (ex) and emission (em) maxima (nm) | Associated component | Possible sources |
|---|---|---|---|
| B | ex 270 - 275, em 304 - 312 | Tyrosine-like (proteinaceous) | Terrestrial, autochthonous production, microbial processing |
| T | ex 270 - 280, em 330 - 368 | Tryptophan-like (proteinaceous) | Terrestrial, autochthonous production, microbial processing |
| M | ex 290 - 325, em 370 - 430 | UVA humic-like. LMW, common in marine environments and associated with biological activity | Terrestrial, autochthonous production, microbial processing |
| A | ex < 260, em 448 - 480 | UVC humic-like. Often HMW and aromatic | Terrestrial |
| C | ex 320 - 360, em 420 - 460 | UVC humic-like. Often HMW and aromatic | Terrestrial |

## 2.6 Dissolved organic carbon concentration

Dissolved organic carbon (DOC) concentration was measured following photodegradation and biodegradation steps to determine the influence of these processes on the total quantity of carbon available. Samples were filtered through a pre-flushed (3 times with 10 mL of Milli-Q) 0.22 μm polyethersulfone (PES) syringe filter (Whatman, England) into acid washed 30 mL HDPE Nalgene bottles and frozen until analysis. DOC concentrations were quantified using a Shimadzu TOC-L Organic Carbon Analyser with a high-sensitivity catalyst. Non-purgeable organic carbon (NPOC) was measured following the acidification of samples with hydrochloric acid and catalytic combustion (680ºC) of DOC to carbon dioxide, which is subsequently measured by infrared absorption. The limit of detection (LoD) was 67 μg L$^{-1}$ with a precision of ± 3 % and an accuracy of ± 2 % as defined by the comparison of a gravimetrically diluted 500 mg L$^{-1}$ TOC certified stock standard to a concentration of 500 μg L$^{-1}$ (Sigma TraceCERT). Procedural blanks were analysed alongside samples to monitor for any contamination which may have been introduced at any stage during the incubation and processing procedures. The DOC concentration decreased by ~ 24 % in the control (Milli-Q inoculated with bacteria) and DOC concentrations in the pigment and surface ice were therefore normalised against the control. The percent of biodegradable or bioavailable DOC



(%BDOC) was calculated as the difference in DOC concentration at the start and end of the 31 day biodegradation
incubation period (Fasching et al., 2014).

**2.7    Bacterial enumeration and biomass**
Bacterial enumeration was conducted at 0, 6, 15 and 30 days from 300 µL of sample (n = 3) by epifluorescence
microscopy following staining with 4', 6-diamidino-2-phenylindole (DAPI, Sigma) at a final concentration of 10
µg mL$^{-1}$ (Porter and Feig, 1980). The staining, filtering and mounting procedure was conducted as outlined by
Bradley *et al.* (2016). Bacterial cells were counted using a Leica DM 2000 epifluorescence microscope at 1000x
magnification with attached MC120 HD microscope camera (Leica, Germany). A minimum of 300 cells or 30
randomly selected grids (each $10^4$ µm$^2$) were counted. Abundance in the pigment and surface ice was normalised
to the control (Milli-Q inoculated with bacteria).

Imaging for the estimation of cell volumes was performed in parallel and measurements of cell diameter and
height made using ImageJ software. Cell volumes were calculated following Eq. (2):
$V = (w^2 * \pi/4) * (l - w) + (\pi * w^3/6),$        (2)
where $V$ (µm$^3$) is the cell volume, and $w$ and $l$ are cell width and length (in µm) (Fasching et al., 2014). Estimated
cell volumes were converted to individual cell carbon content according to Bratbak and Dundas (1984). Microbial
biomass was then calculated as the product of bacterial abundance and the individual cell carbon content for each
sample. Bacterial growth efficiency (BGE) is an indicator of the use of DOM for bacterial growth and gives an
indication of the flow of carbon through the bacterial biomass (Anesio et al., 2005; Del Giorgio and Cole, 1998).
BGE was estimated as the change in biomass divided by the change in DOC concentration (assumed to represent
the DOC incorporated into the bacterial biomass plus respiration) over the 31-day incubation period. This assumes
carbon incorporated into the bacterial biomass was not utilised for respiration.

**2.8    Data analysis**
All statistical analyses and plotting of data were performed using R v.3.4.1 (R Core Team, 2019). Prior to
parametric analysis of datasets, Shapiro-Wilks Test combined with interrogation of frequency histograms was
used to determine normality. Three-way analysis of variance (ANOVA) with the fixed factors of carbon source
(i.e. pigment or surface ice; 2 levels); light (4 levels) and degradation type (i.e. photodegradation or
biodegradation; 2 levels) was used to determine significant differences in the absorbance and fluorescence indices.
Principal Component Analysis (PCA) was also used to summarise normalised and centred absorbance and
fluorescence indices utilising the 'factoextra' R package (Kassambara, 2020). Differences within DOC
concentration and bacterial abundance were described using a three-way ANOVA with the fixed factors of 'time'
(4 levels), 'light' (4 levels), nutrients (i.e. normal or +nutrients; 2 levels) for the pigment and surface ice.



## 3    Results

### 3.1    DOM composition

#### 3.1.1    UV-Vis absorbance properties

Glacier algal pigment DOM that was not exposed to radiation (DARK) displayed two prominent peaks in absorption at $\lambda_{285nm}$ and $\lambda_{304nm}$ as well as a secondary peak at $\lambda_{385nm}$ (Figure 1). These peaks were also evident in pigment exposed to PAR and UV+PAR, which exhibited spectra highly comparable to DARK. In contrast, UV-irradiated pigment revealed marked differences in absorption with a conspicuous depression in the peaks at $\lambda_{285nm}$ and $\lambda_{304nm}$ and 10 % greater absorption at $\lambda_{250nm}$ and between $\lambda_{325 - 600nm}$. This was reflected in average specific UV absorbance (SUVA) indices (Table 3) for UV-irradiated pigment which were 0.9 L m$^{-1}$ mg$^{-1}$ lower than DARK for SUVA$_{280}$ and between 0.8 - 1.0 L m$^{-1}$ mg$^{-1}$ higher for SUVA$_{254}$ and SUVA$_{365}$. For all light treatments, absorption decreased with increasing wavelength across visible wavelengths (400 – 700 nm). Average absorption at 440 nm (A$_{440}$) was $16.9 \pm 0.7$ m$^{-1}$ in DARK and did not differ significantly across light treatments.

The inoculation of DARK pigment with bacteria (biodegradation) resulted in pronounced differences in the absorbance spectra. For example, a conspicuous 33 % depression was evident in the peaks at $\lambda_{285nm}$ and $\lambda_{304nm}$ and the peak at $\lambda_{385nm}$ was absent. In addition, a secondary peak at $\lambda_{330nm}$ developed which was not evident in the initial DARK spectra. SUVA$_{254}$ and SUVA$_{365}$ indices were significantly higher following biodegradation whereas SUVA$_{280}$ was significantly lower ($F_{3,32} = 21.8$, $p < 0.001$; $F_{3,32} = 20.0$, $p < 0.001$; $F_{3,32} = 36.3$, $p < 0.001$ respectively). Spectral differences across light treatments were also apparent following biodegradation with the largest deviations in absorption occurring within UV wavelengths (200 – 400 nm). In particular, UV-irradiated pigment retained the greatest absorption at $\lambda_{285nm}$ and $\lambda_{304nm}$ therefore SUVA$_{280}$ for this treatment was significantly larger than DARK ($F_{3,32} = 1.7$, $p < 0.01$). All light treatments exhibited an ~ 20 % increase in A$_{440}$ following biodegradation.

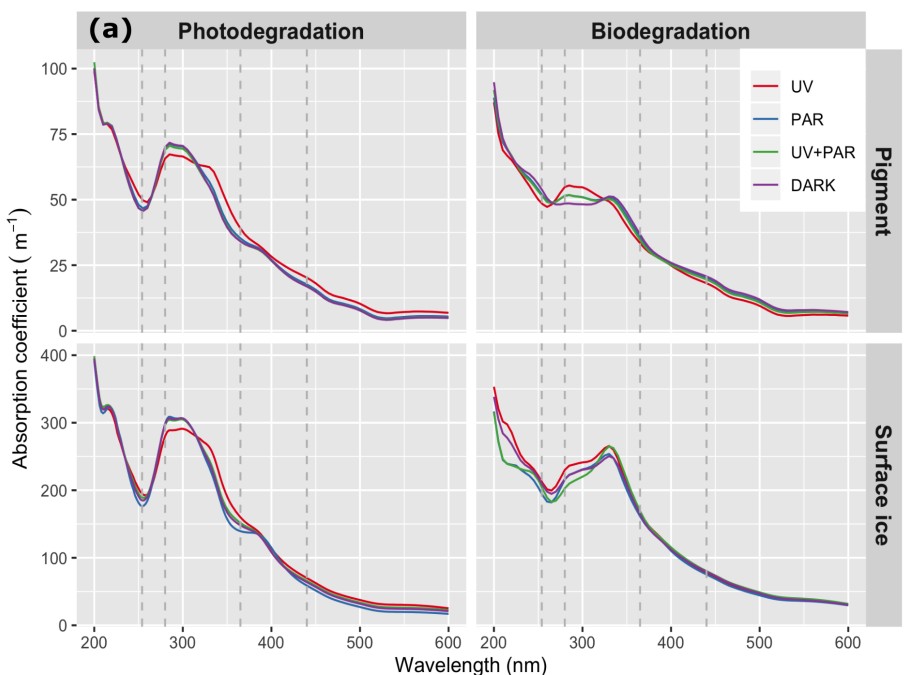

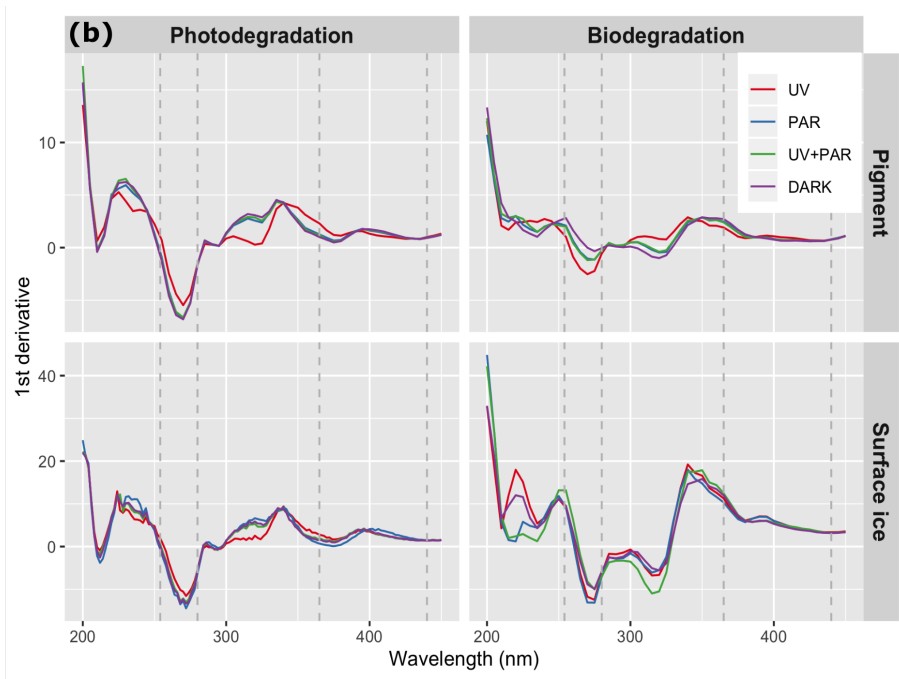

275
276

**Figure 1: Average (a) and first derivative (b) of absorbance spectra across light treatments for the pigment and surface ice after exposure to light regimes (photodegradation) and following 31 days incubation with bacteria (biodegradation) (n=3). Dashed lines indicate the wavelengths at which specific UV absorbance (SUVA) was calculated (254 nm, 280 nm, 365 nm) and the absorbance at 440 nm (A$_{440}$).**





**Table 3: Specific UV absorbance values (L m⁻¹ mg⁻¹) at 254 nm (SUVA254), 280 nm (SUVA280) and 365 nm (SUVA365)**
**following photodegradation (Photo) and biodegradation (Bio) for the pigment and surface ice (mean ± SE, n=3).**
**Highlighted values reflect trends on the absorbance spectra outlined previously. No significant differences were**
**identified across light treatments per degradation type (i.e. photodegradation or biodegradation) per carbon source**
**(pigment or surface ice), apart from SUVA280 indices for pigment. Letters to denote homogenous subsets (lower case)**
**are only displayed for this exception. Upper case letter denote significant differences between photodegradation and**
**biodegradation per light treatment per carbon source.**

| DOM source | Light treatment | SUVA$_{254}$ | | SUVA$_{280}$ | | SUVA$_{365}$ | |
|---|---|---|---|---|---|---|---|
| | | Photo | Bio | Photo | Bio | Photo | Bio |
| Pigment | UV | **9.8 ± 0.2$^A$** | 12.5 ± 0.7$^A$ | **12.9 ± 0.2$^A$** | $^a$14.2 ± 0.7$^A$ | **7.7 ± 0.1$^A$** | 8.6 ± 0.7$^B$ |
| | PAR | 9.2 ± 0.1$^A$ | 13.4 ± 0.4$^A$ | 13.8 ± 0.1$^A$ | $^{ac}$13.4 ± 0.3$^A$ | 7.0 ± 0.1$^A$ | 9.2 ± 0.5$^B$ |
| | UV + PAR | 9.1 ± 0.3$^A$ | 10.9 ± 0.3$^A$ | 13.7 ± 0.3$^A$ | $^{bc}$11.0 ± 0.1$^B$ | 6.8 ± 0.4$^A$ | 7.4 ± 0.3$^B$ |
| | Dark | 9.0 ± 0.2$^A$ | 12.9 ± 0.4$^A$ | 13.8 ± 0.2$^A$ | $^c$11.7 ± 0.1$^B$ | 6.7 ± 0.2$^A$ | 8.9 ± 0.6$^B$ |
| Surface ice | UV | **9.0 ± 0.4$^A$** | 13.2 ± 0.2$^B$ | **12.9 ± 0.6$^A$** | 14.5 ± 0.3$^A$ | **7.4 ± 0.5$^A$** | 10.5 ± 0.3$^A$ |
| | PAR | **8.1 ± 0.6$^A$** | 12.8 ± 0.3$^B$ | 13.7 ± 0.6$^A$ | 14.4 ± 0.3$^A$ | 6.4 ± 0.7$^A$ | 10.7 ± 0.3$^B$ |
| | UV + PAR | 8.7 ± 0.1$^A$ | 13.3 ± 0.4$^B$ | 13.6 ± 0.1$^A$ | 13.3 ± 0.2$^A$ | 7.0 ± 0.2$^A$ | 11.1 ± 0.4$^A$ |
| | Dark | 8.4 ± 0.2$^A$ | 14.1 ± 0.1$^B$ | 13.8 ± 0.2$^A$ | 14.8 ± 0.8$^A$ | 6.8 ± 0.2$^A$ | 11.2 ± 0.3$^A$ |


DARK surface ice DOM exhibited absorption on average ~ 4 times greater than the DARK pigment DOM (Figure
1); however, distinct similarities in absorption were evident, particularly the corresponding peaks at $\lambda_{285nm}$, $\lambda_{304nm}$
and $\lambda_{385nm}$. The average $A_{440}$ was 62.6 ± 2.4 m⁻¹ which was 3 times larger than the DARK pigment. Photodegraded
surface ice exhibited spectral differences, with the most striking disparities in the UV and PAR treatments. In
particular, surface ice exposed to UV radiation exhibited ~ 5 % less absorption at $\lambda_{304nm}$ and an average SUVA$_{280}$
1.1 L m⁻¹ mg⁻¹ lower than DARK (Table 3). Equally, PAR-irradiated surface ice exhibited lower absorption at
$\lambda_{250nm}$ and $\lambda_{350 - 370nm}$ and subsequently the average SUVA$_{254}$ was 5 % lower than DARK.

The biodegraded DARK surface ice absorption spectra exhibited similar trends to the pigment with a ~ 20 %
depression in peaks at $\lambda_{285nm}$ and $\lambda_{304nm}$; however, retained more defined peaks at these wavelengths compared to
the DARK pigment. In addition, a much more pronounced secondary peak at $\lambda_{330nm}$ was visible in the DARK
surface ice. Although SUVA$_{254}$ in the DARK treatment increased significantly by 65 % following biodegradation
($F_{3,32}$ = 21.8, p < 0.001), the other SUVA indices were not significantly different. Deviations in the spectra across
light treatments is largely confined between $\lambda_{200 - 350nm}$, with UV exhibiting the largest absorption over these
wavelengths and UV and UV+PAR the lowest. Absorbance across visible wavelengths was highly consistent
across light treatments and a ~ 20 % increase $A_{440}$ was evident following biodegradation.




### 3.1.2    Fluorescence properties

Peaks commonly identified in the fluorescence spectra of natural waters (B, T, M, A, C) (Coble, 1996; Coble et
al., 2014) were all present in the DARK pigment treatment (Figure 2; Figure 3). We observed an approximate
50:50 ratio between peaks associated with fluorophores in proteinaceous (B and T) and humic-like (A, C and M)
DOM in the DARK treatment. Following photodegradation of the pigment, the relative intensity of all peaks was
comparable with DARK and only a slight increase in peak A was observed in the PAR treatment. Biodegradation
was observed to alter the fluorescence signature of the pigment with humic-like fluorescence increasing by almost
25 % in the DARK treatment. UV and UV+PAR-irradiated pigment resulted in very little difference in
fluorescence intensities compared to DARK. Only the PAR treatment exhibited noticeable changes with ~ 10 %
greater fluorescence of peak B and a similar decrease in peak A compared to DARK.
In contrast to the pigment, DARK surface ice was dominated by fluorescence associated with humic-like DOM
(> 75 %) consisting predominantly of peaks A and C. Following photodegradation, the relative intensity of peaks
deviated between light treatments. For example, the fluorescence intensities of humic-like peaks (A and C) was ~
12 % lower in the UV treatment compared to DARK. In addition, PAR and UV+PAR exposed surface ice did not
contain peak T fluorescence but exhibited almost double the fluorescence associated with peak M. The
biodegradation of surface ice resulted in an ~ 10 % increase in fluorescence associated with humic-like DOM in
the DARK treatment. Reductions in proteinaceous fluorescence in DARK was particularly evident for peak B
which was almost undetectable following biodegradation (Figure 3). Overall, PAR and UV treatments retained
the largest proportion of proteinaceous peaks, predominantly due to increased fluorescence of peak T and M.

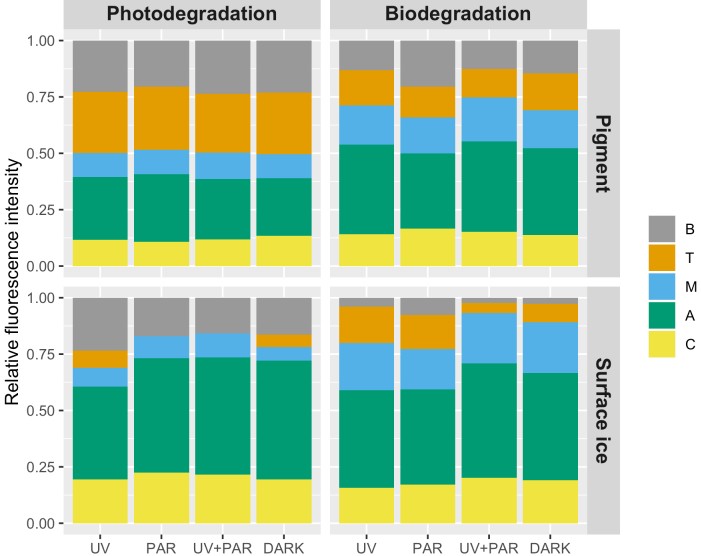

**Figure 2: The mean relative fluorescence intensity (R.U.; n = 3) associated with commonly identified peaks (B, T, M, A and C) in the fluorescence spectra of natural waters (Coble, 1996; Coble et al., 2014) (n=3). Peaks B and T are often associated with protein**
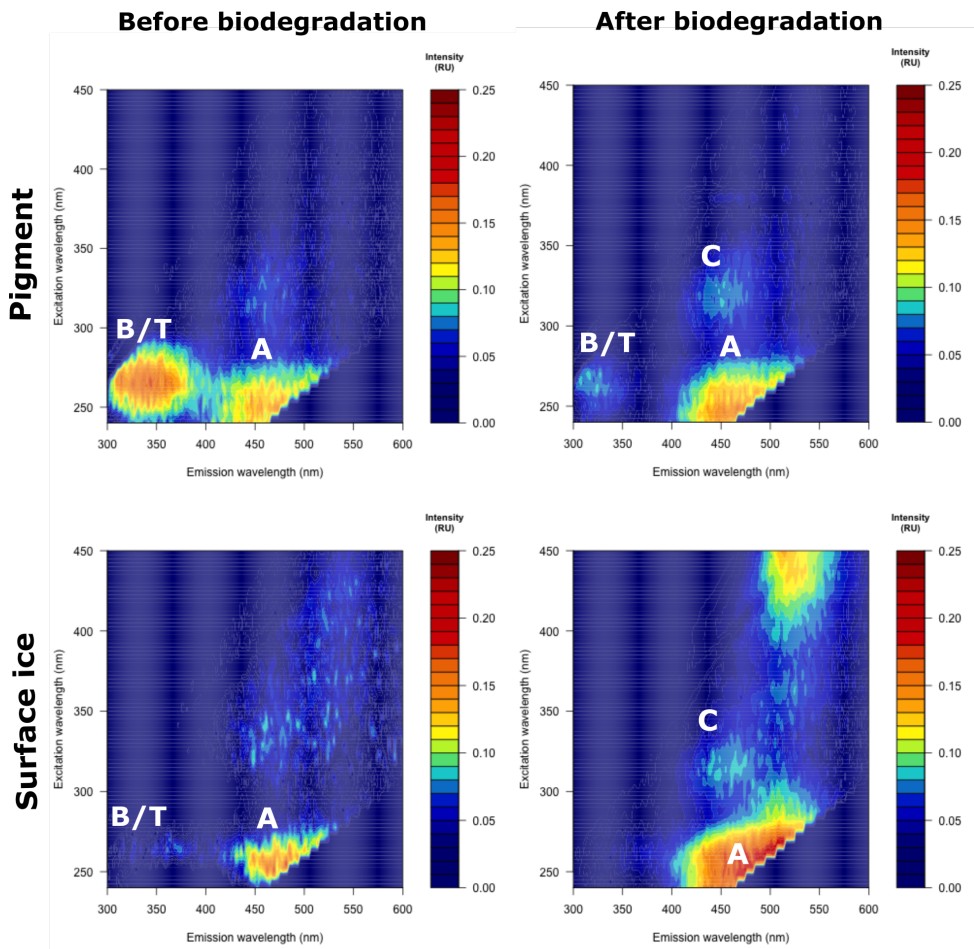

**Figure 3: Excitation-Emission matrices (EEMs) for the DARK pigment and surface ice before and after biodegradation. Peaks identified are those typically identified in natural water fluorescence spectra (Birdwell and Engel, 2010; Coble, 1996; Coble et al., 2014). Peaks A and C are generally associated with fluorophores in humic-like DOM whereas peaks B and T are associated with proteinaceous DOM.**

The average degree of humification (HIX) across all light treatments in the pigment was $0.51 \pm 0.01$ which was significantly lower than the surface ice average HIX of $0.57 \pm 0.11$ ($F_{1,32} = 46.8$, $p < 0.001$). Following photodegradation, HIX was not significantly different across light treatments for either pigment or surface ice (Figure 4). However, HIX increased significantly by 29 % and 12 % in the pigment and surface ice respectively following biodegradation (pigment: $F_{1,32} = 46.8$, $p < 0.001$; surface ice $F_{1,32} = 46.8$, $p < 0.01$). The average fluorescence index (FI) was not significantly different between the pigment ($0.51 \pm 0.15$) and surface ice ($0.57 \pm 0.11$). Following biodegradation, FI of the pigment increased significantly ($F_{1,32} = 3.5$, $p < 0.05$) whereas surface ice FI remained relatively constant.


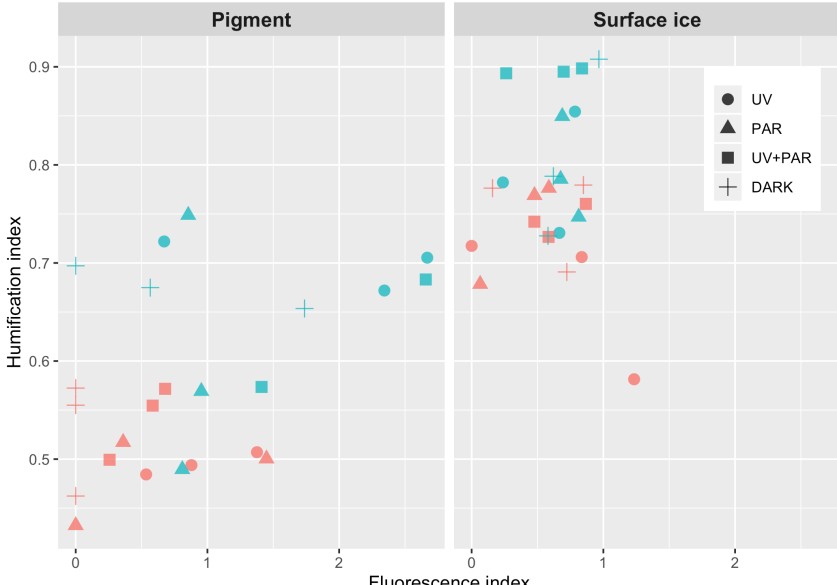

**Figure 4: Humification index (HIX) against the fluorescence index (FI) for pigment and surface ice after photodegradation (red) and after biodegradation (blue) across all light treatments. Increasing HIX indicates the presence of more humic compounds. Low FI has been associated with terrestrially derived DOM and high FI with microbial DOM.**

Principal component analysis (PCA) was utilised to summarise all absorbance and fluorescence indices and to elucidate underlying trends of the dataset (Figure 5; Table 4). PC1 described 43 % of the variance in the data and shows strong positive loadings for parameters associated with high molecular weight, aromatic compounds (peak M fluorescence intensity, $SUVA_{365}$ and peak A intensity; Table 4). PC2 accounted for 23 % of the variance and represented Specific visible absorbance at $\lambda 440$ ($SVA_{440}$), spectral slope between 300 - 700 nm ($S_{300-700}$) and $SUVA_{254}$. There were no significant differences in PC1 or PC2 between light treatments; however, carbon source (i.e. pigment or surface ice) and type of degradation (i.e. photodegradation or biodegradation) were found to be significant drivers of change in PC1 ($F_{1,46} = 31.8$, $p < 0.001$; $F_{1,46} = 35.6$, $p < 0.001$ respectively) and PC2 ($F_{1,46} = 48.7$, $p < 0.001$; $F_{1,46} = 18.9$, $p < 0.001$ respectively).





**Figure 5: Principle component analysis (PCA) of the absorbance and fluorescence for photodegraded (yellow) or biodegraded (green) DOM. Points representing pigment DOM are circled in red and the remaining points represent surface ice DOM. PC1 accounts for 43 % and PC2 accounts for 23 % of the variability in the dataset. Absorbance indices include SUVA254, SUVA280, SUVA365, A440, spectral slope between 300 – 700 nm (S300_700) and fluorescence indices include fluorescence intensity of peaks B, T, M, A, C, humification index (HIX) and fluorescence index (FI). Boxplots of PC1 against carbon source (a), light (b) and degradation type (i.e. photodegradation or biodegradation; c) and PC2 against carbon source (d), light (e) and degradation type (f). Letters denote significant differences between groups (p < 0.05)**





**Table 4: Component loadings of PCA analysis of absorbance and fluorescence indices. The largest component loading**
**is highlighted in grey. Absorbance indices include SUVA254, SUVA280, SUVA365, A440, spectral slope between 300 –**
**700 nm (S300_700) and fluorescence indices include fluorescence intensity of peaks B, T, M, A, C, humification index**
**(HIX) and fluorescence index (FI).**

|  | PC1 | PC2 | PC3 |
|---|---|---|---|
| Proportion of variance | 42.7 % | 22.7 % | 10.1 % |
| Cumulative proportion | 42.7 % | 65.4 % | 75.5 % |
| **Component loadings** |  |  |  |
| B | -0.12 | -0.26 | 0.28 |
| T | 0.10 | 0.03 | 0.63 |
| M | 0.39 | -0.07 | 0.07 |
| A | 0.35 | -0.31 | -0.04 |
| C | 0.33 | -0.34 | -0.01 |
| FI | -0.04 | 0.20 | -0.51 |
| HIX | 0.33 | -0.13 | -0.45 |
| $S_{300-700}$ | -0.24 | -0.39 | -0.09 |
| $SUVA_{254}$ | 0.31 | 0.36 | 0.05 |
| $SUVA_{280}$ | 0.12 | -0.25 | 0.18 |
| $SUVA_{365}$ | 0.37 | 0.22 | 0.09 |
| $SVA_{440}$ | 0.28 | 0.40 | 0.07 |
| $A_{440}$ | 0.33 | -0.34 | -0.07 |


### 3.2  DOC quantity

Average dissolved organic carbon (DOC) concentration of the DARK pigment was 2.5 ± 0.04 mg $L^{-1}$ and
photodegradation did not significantly alter concentrations in the UV, PAR or UV+PAR treatments. During the
31-days of biodegradation, DOC concentrations in the DARK treatment exhibited high levels of variability and
by the end of the incubation period had decreased by ~ 0.25 mg $L^{-1}$, but this was not significant (Figure 6).
Concentrations across light treatments were equally variable decreasing by ~ 1 mg $L^{-1}$ over 31 days; however,
were not significantly different from DARK. The percent of biodegradable DOC (%BDOC) was ~ 10 % and was
not significantly different between UV and PAR exposed and DARK pigment (Appendix A; Table A1). The
addition of nutrients (+Nutrients) to incubations did not affect the DOC concentration across any of the light
treatments; however, overall %BDOC in +Nutrients was significantly higher than the normal incubations ($F_{1,61}$ =
8.7, p < 0.01; Appendix A).



The average DOC concentration ($19.1 \pm 0.03$ mg L$^{-1}$) in the DARK surface ice was ~ 8 times greater than in the
pigment and no significant change in concentration was found across light treatments following photodegradation.
Nevertheless, DOC concentrations did decrease significantly during biodegradation incubations across all light
treatments ($F_{5,212} = 432$, $p < 0.001$). DOC concentrations in DARK were significantly lower than in other light
treatments at 6 days ($F_{3,36} = 41.4$, $p < 0.01$ for all) and 10 days ($F_{3,36} = 32.9$, $p < 0.001$ for all). The UV, PAR and
UV+PAR treatments followed similar trajectories during the incubation period; however, only the DOC
concentration in the UV treatment was significantly higher than in DARK at 31 days ($F_{3,36} = 4.7$, $p < 0.01$). The
BDOC of surface ice was ~ 30 % and was significantly higher than the pigment ($F_{1,61} = 839$, $p < 0.001$); however,
there was no significant difference across light treatments (Appendix A). The DOC concentration in the normal
treatment was not significantly different from the +Nutrients treatments following biodegradation.

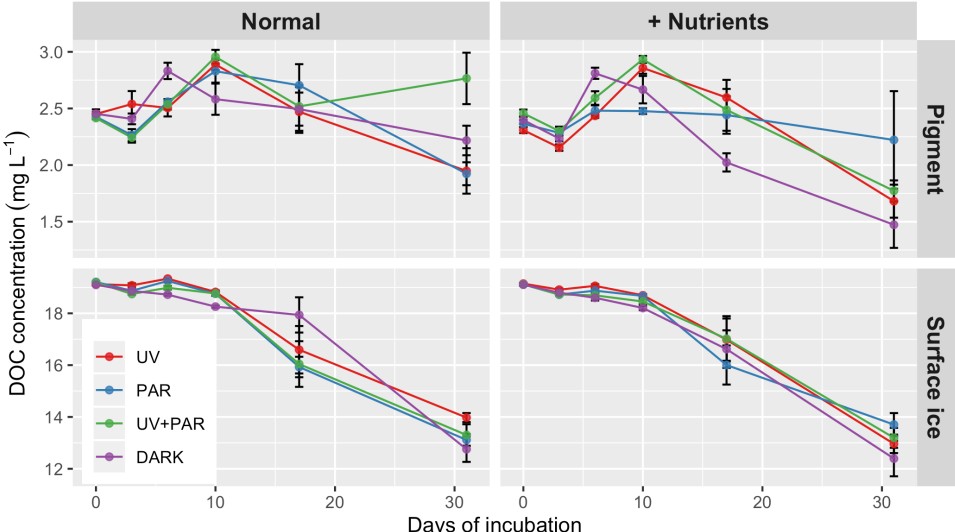


**Figure 6: Dissolved organic carbon (DOC) concentration (mg L-1) during the 31-day biodegradation period in the**
**normal and nutrient addition (+Nutrients) incubations for the pigment and surface ice (mean ± SE, n = 5).**

### 3.3    Bacterial abundance and growth efficiency

Bacterial abundance in pigment incubations did not increase significantly over the 31-day incubation period across
any of the light treatments (Figure 7). Indeed, when normalised against the change in bacterial abundance in the
Milli-Q control, abundance actually decreased in the PAR, UV+PAR and DARK treatments during the incubation
period. In the +Nutrients treatment, abundance increased in the UV, PAR and DARK treatments to $0.7 \pm 0.2$ x$10^6$
cells mL$^{-1}$; however, this was not significant. Average bacterial growth efficiency (BGE) for all pigment
treatments was < 4 % and substantial variability was observed across light and nutrient treatments (Appendix A).

In contrast to the pigment incubations, bacterial abundance across all surface ice treatments increased significantly
between 3 and 9-fold during the incubation period ($F_{3,61} = 18.4$, $p < 0.001$). At 17 days, the largest increases were
observed in the UV and UV+PAR treatments which supported an abundance of $2.0 \pm 1.1$ x$10^6$ cells mL$^{-1}$ and 1.7





± 0.3 x10⁶ cells mL⁻¹ respectively. Rapid growth in PAR-irradiated surface ice was observed in the final 14 days
and as such the final abundance of 9.0 ± 1.7 x10⁶ cells mL⁻¹ was significantly larger than DARK ($F_{3,61}$ = 2.2, p <
0.05). Average abundance at the end of biodegradation across all light treatments was 4.7 ± 1.5 x10⁶ cells mL⁻¹
and was not significantly different in the +Nutrients treatment. Average BGE in the surface ice was 6.7 ± 1.3 %
which was significantly higher than the pigment ($F_{1,31}$ = 4.7, p < 0.05). However, BGE was not significantly
different across light or nutrient treatments (Appendix A).

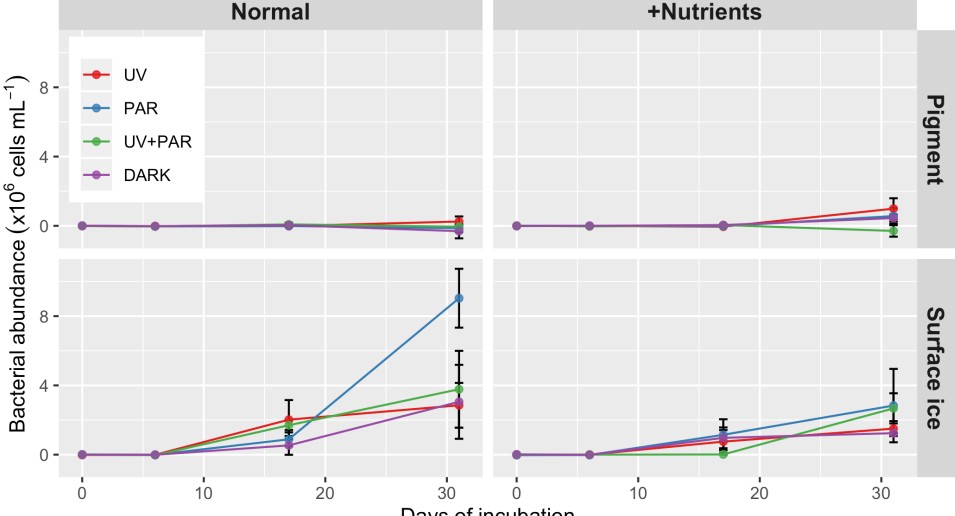

**Figure 7: Bacterial abundance in pigment and surface ice coloured by light treatment across the nutrient and nutrient**
**addition (+Nutrients) incubations (mean ± SE, n = 3)**

## 4   Discussion

Active microbial communities store and transform carbon across the Greenland Ice Sheet (GrIS) supraglacial
environment. Glacier algae residing within the surface ice contain a dark coloured photoprotective pigment, which
comprises a large proportion of the cell (~ 4 % of the dry weight) (Remias et al., 2012; Williamson et al., 2020),
and may thus represent a vast source of carbon. Glacier algae are responsible for the majority of carbon fixation
within the surface ice (Williamson et al., 2018; Yallop et al., 2012), which is an essential autochthonous carbon
source for heterotrophic bacterial communities (Nicholes et al., 2019; Yallop et al., 2012). Despite the surface ice
receiving extremely high levels of irradiation, the role of photodegradation on carbon flows was yet to be
constrained. This study assessed responses in the composition and quantity of glacier algae secondary pigment
and surface ice DOM sources following exposure to UV, PAR, UV+PAR (photodegradation) and subsequent
incubation with bacterial communities isolated from the ambient environment (biodegradation). Our results
indicate that the composition of algal pigment and surface ice DOM is altered following exposure to radiation,
but that the quantity of DOC remains constant. Biodegradation caused the greatest changes to DOM composition
and DOC quantity, particularly for surface ice DOM sources.





### 4.1 Glacier algal phenolic pigment


The secondary phenolic pigment extracted from glacier algae exhibited absorbance that was highly comparable
with previous characterisations of this substance, displaying strong absorption particularly over UV-A and UV-B
wavelengths, which decreased across the visible spectrum (Remias et al., 2012; Williamson et al., 2020). Peaks
in absorption observed at $\lambda_{285nm}$, $\lambda_{304nm}$ and $\lambda_{385nm}$ likely reflect different moieties within the pigment structure
(Remias et al., 2012; Williamson et al., 2020). For example, peaks at $\lambda_{304nm}$ and $\lambda_{389nm}$ have previously been
identified as purpurogallin carboxylic acid-6-$O$-β-$_D$-glucopyranoside ($C_{18}H_{18}O_{12}$), which is formed from the
chemical oxidation of gallic acid or from a mixture of gallic acid and pyrogallol (Polewski et al., 2002; Remias et
al., 2012). Equally, a peak at $\lambda_{278nm}$ is thought to be gallic acid glycoside and may represent an important
biosynthetic precursor to purpurogallin carboxylic acid (Remias et al., 2012). Given the similarity in absorption
peaks, it is likely these compounds are also present in the pigment utilised for this experiment. The chemical
composition and absorption of pigment likely reflects its primary role of protecting chloroplasts from damaging
UV and high energy blue visible radiation, while transmitting longer, less damaging wavelengths for
photosynthesis (Williamson et al., 2020).

We observed structural changes to light-sensitive (chromophoric) regions of the glacier algal pigment following
exposure to UV radiation; however, the quantity and bioavailability of compounds remained consistent across
light treatments. UV-irradiated pigment exhibited a depression in absorption associated with purpurogallin
carboxylic acid and gallic acid glycoside, suggesting that these compounds were subject to photodegradation.
Ward and Cory (2016) highlighted that carboxylic acids in Arctic algal mats and permafrost were highly
susceptible to photodegradation and form a variety of hydrocarbons. The concomitant increase in $SUVA_{254}$ and
$SUVA_{365}$ may indicate the transformation into aromatic compounds, which absorb at $\lambda_{254nm}$ and $\lambda_{365nm}$ (Uyguner
and Bekbolet, 2005; Weishaar et al., 2003). Alternatively, UV radiation may preferentially degrade aliphatic, low
molecular weight (LMW) DOM resulting in a greater proportion of aromatic DOM (Ward and Cory, 2016). This
is common in other aquatic environments and results in UV radiation effectively reducing the availability of LMW
DOM sources to heterotrophic communities (Amado et al., 2015; Ward and Cory, 2016). Despite compositional
changes to pigment DOM following UV exposure, bulk DOM quantity remained constant across light treatments.
Photosensitive DOM (i.e. CDOM) accounts for only a fraction of the total DOM pool (Fleck et al., 2014), therefore
structural alterations can occur without altering the bulk DOM quantity (Cory et al., 2011; Spencer et al., 2007).
Given that pigment DOM fluorescence also remained consistent across light treatments, our data are consistent
with variation in pigment DOM composition but not quantity following exposure to potential photodegradation.

Biodegradation incubations revealed that certain components of glacier algal pigment may be transformed by the
heterotrophic bacterial community. Consistent with previous studies, $SUVA_{254}$ and $SUVA_{365}$ increased across
incubations, concomitant with decreasing contributions of proteinaceous fluorescence and suggestive of
preferential consumption of LMW aliphatic DOM by bacterial communities (Antony et al., 2018; Hansen et al.,
2016; Kirchman, 2003). Most notably, reduced absorption associated with purpurogallin carboxylic acid and
gallic acid glycoside indicates that these compounds may comprise a proportion of the bioavailable substrates
consumed by the bacterial community. Carboxylic acids have been observed to be readily assimilated into
bacterial biomass, thus these compounds are considered largely bioavailable (Bertilsson and Tranvik, 1998).



Interestingly, irradiated pigment retained greater purpurogallin carboxylic acid and gallic acid glycoside
absorption compared to DARK treatments, highlighting that UV and PAR exposure either reduces the
bioavailability of these compounds or produces more bioavailable compounds that are preferentially consumed
(Amado et al., 2015; Lindell et al., 1995; Zepp and Moran, 1997). The humificiation (HIX) of pigment and the
dominance of peak M indicated the production of LMW, humic DOM that is widely attributed to biological
activity (Coble, 1996; Murphy et al., 2008). This is corroborated by the increased FI following biodegradation
confirming a greater dominance of DOM of a microbial origin (McKnight et al., 2001). Overall, biodegradation
incubations revealed the potential for bacterial communities to transform a small proportion of bioavailable carbon
(~ 10 %) sourced from glacier algal secondary pigment and demonstrated that UV degradation may influence
which DOM components are degraded.

Despite the marked changes in DOM composition following biodegradation, our results indicated that only ~ 3 %
of carbon was incorporated into bacterial biomass and bacterial abundance thus remained relatively constant
across all light treatments. This suggests that glacier algal pigment may be a low-quality carbon substrate for
surface ice bacterial communities and the overall quality of carbon was not changed by photodegradation. We
provide two explanations for this, both related to the polyphenolic nature of purpurogallin carboxylic acid and
gallic acid glycoside. Polyphenols represent a variety of chemical substances found in algae and higher plants that
have a range of ecological and physiological functions (Bhat et al., 1998; Cannell et al., 1988). Notably,
polyphenols have antimicrobial properties and play an essential role in protecting cells against bacterial and fungal
pathogens (Cannell et al., 1988; Lima et al., 2016; Nguyen et al., 2013; Scalbert, 1991). A range of bacterial
species are susceptible to polyphenol inhibition (Scalbert, 1991; Taguri et al., 2006) and we propose that the
majority of the surface ice bacterial community are affected. Despite this, compositional changes to DOM outlined
previously may suggest that some resistant species reside within the surface ice environment. Bacterial
degradation or modification of polyphenols, as observed in our incubations, has been highlighted as an important
mechanism for overcoming inhibition, such as the ability of *Achromobacter sp.* to grow in a gallotannin media
(Bhat et al., 1998; Scalbert, 1991; Smith et al., 2005). It is possible that the inhibition of the majority of the
bacterial community by the polyphenolic nature of pigment may have resulted in low bacterial growth efficiency.

In addition to the potential inhibition of bacteria by antimicrobial properties of glacier algal phenolic pigment,
nutrient limitation may have also restricted bacterial growth. Nitrogen and phosphorus are essential
macronutrients obtained by bacteria from both inorganic and organic sources (Dodds, 2010; Sigee, 2004). Organic
sources include proteins, nucleic acids and amino acids, which are actively and passively released from cells via
extrapolymeric substances and cell lysis (Sigee, 2004). Both purpurogallin carboxylic acid and gallic acid
glycoside do not contain nitrogen or phosphorus and hence are easily synthesised in high light, low nitrogen
environments, such as supraglacial surface ice (Remias et al., 2009, 2012). Accordingly, organic nutrient sources
were likely very limited in the pigment incubations, potentially restricting bacterial activity, driving utilisation of
~ 20 % of carbon. In contrast, incubations spiked with nitrogen and phosphorus (+Nutrients) exhibited much
greater bacterial growth and a ~ 10 % increased BDOC, thus additional nutrients likely facilitated a greater
exploitation of carbon sources.





Although glacier algal secondary pigment could represent a substantial carbon source within the surface ice during
bloom events, the mechanism of pigment release remains unconstrained. Given the metabolic cost of producing
this secondary pigment and its essential role protecting the cell from photodamage, it is unlikely to be actively
excreted by glacier algae. Alternative mechanisms of release may include passive leakage or lysis of cells due to
natural senescence or viral/fungal attack (Dodds, 2010; Sigee, 2004; Zlotnik and Dubinsky, 1989); however, the
degree of leakage and mortality rate of glacier algae remains unconstrained. Overall, biodegradation incubations
revealed that glacier algal phenolic pigment is largely unavailable to heterotrophic bacterial communities from
the surface ice of the GrIS. Despite this, algal pigment may represent a viable carbon source for Archaea or fungi
within the surface ice community and further investigation is therefore required to reveal the fate of this carbon
source.

**4.2    Surface ice DOM**
To the best of our knowledge, this is the first spectroscopic characterisation of surface ice DOM from the Dark
Zone of the Greenland Ice Sheet. Our results highlight remarkable similarities in absorption characteristics
between surface ice and pigment DOM indicating the presence of purpurogallin carboxylic acid and gallic acid
glycoside in surface ice. This may be a result of natural glacier algal cell lysis; however, we also acknowledge
that algal cells may have lysed during experimental set up, releasing pigment and thus further *in situ* investigations
are required to clarify whether this is representative. Despite this, purpurogallin carboxylic acid and gallic acid
glycoside were some of the most strongly light absorbing compounds, therefore the release of even small
concentrations of pigment to the ambient environment could substantially alter the optical properties of surface
meltwater. Surface ice DOM exhibited predominantly humic-like fluorescence dominated by peaks A and C; both
of which are often associated with the presence of HMW, aromatic DOM of terrestrial plant or soil origin (Coble,
1996; Coble et al., 1990; Fellman et al., 2010). This is surprising given the high levels of primary production by
glacier algae reported from the surface ice (Williamson et al., 2018; Yallop et al., 2012) and the relatively low
input of carbon from allochthonous sources in our sampling region on the GrIS (Stibal et al., 2012). However,
peaks A and C have also been observed following bacterial degradation of autochthonous DOM (Stedmon and
Markager, 2005) and it is therefore possible that surface ice DOM had already undergone a degree of
biodegradation prior to sampling. Equally, the formation of humic-like DOM following photodegradation of algal-
derived DOM is widely reported (Amado et al., 2015; Stefan et al., 2000; Tranvik and Kokalj, 1998). Carbon
transformation and cycling within the surface ice is thus highly dynamic, with rapid production and consumption
of bioavailable compounds via biotic and abiotic processes.

The exposure of surface ice to UV and PAR radiation resulted in a number of compositional changes to DOM.
Purpurogallin carboxylic acid and gallic acid glycoside were susceptible to degradation on exposure to UV
radiation and responsible for the major shifts in absorption and SUVA indices. An increased proteinaceous
signature dominated by peak B in UV-irradiated surface ice indicated that UV radiation degraded high molecular
weight (HMW) humic DOM into smaller, more bioavailable compounds. This is widely reported from other high-
light aquatic environments and was found to stimulate bacterial production and growth (Amado et al., 2015;
Anesio et al., 2005; Lindell et al., 1995; Zepp and Moran, 1997). Along with UV radiation, PAR was also found





to alter the chemical structure of surface ice DOM, with decreased $SUVA_{254}$ and $SUVA_{365}$ representing a lower
proportion of aromatic DOM compared to DARK treatment samples. A greater proportion of humic- like
fluorescence, characterised by a lack of peak T and a dominant peak M, in PAR- and UV+PAR- irradiated surface
ice suggested that proteinaceous DOM is converted to LMW, humic-like DOM by PAR. It is likely that in the
UV+PAR treatment, these opposing processes (the degradation and formation of humic-like DOM) are occurring
simultaneously, with formation driven by PAR representing the dominant process. This contrasts with the findings
of Antony *et al.*, (2018); however, may simply represent the difference in DOM composition between surface ice
of the GrIS and snow in Antarctica. Although DOC concentrations were consistent across light treatments, we
have demonstrated that surface ice DOM undergoes structural and compositional changes following exposure to
UV and PAR.

Biodegradation incubations revealed that the heterotrophic bacteria community was able to extensively rework
surface ice DOM. The DARK surface ice exhibited increased SUVA indices, confirming that bacteria primarily
consume aliphatic, LMW compounds (Antony et al., 2018; Hansen et al., 2016; Kirchman, 2003). This was further
corroborated by an increase in peak M fluorescence and humification (HIX), highlighting preferential bacterial
consumption of proteinaceous surface ice DOM and production of LMW humic-like DOM (Murphy et al., 2008).
Although purpurogallin carboxylic acid and gallic acid glycoside were degraded across all treatments, absorption
was still evident demonstrating that these compounds were less rigorously degraded than in the pigment
incubations. This may indicate that degradation of these compounds is metabolically intensive and comparatively
less attractive in the presence of alternative carbon sources. The degradation of these compounds within surface
ice DOM supported substantial increases in bacterial abundance throughout incubations. It is likely that the greater
diversity of DOM compounds in surface ice provided a plethora of substrates that facilitated growth across a
broader range of bacterial species within the community (Antony et al., 2017; Smith et al., 2018). BGE in surface
ice was highly comparable with that in glacier forefields (Bradley et al., 2016) and 3-times higher than within
supraglacial streams (Foreman et al., 2013). Bacteria are thus capable of assimilating a greater proportion of
bioavailable carbon from the surface ice compared to supraglacial streams.

Following the biodegradation of photodegraded surface ice, substantial variability in absorbance and fluorescence
across light treatments was observed, highlighting the impact of photodegradation on bacterial DOM
consumption. For example, UV-irradiated surface ice retained the greatest proportion of aromatic DOM, whereas
PAR and UV+PAR retained the least. Additionally, UV- and PAR-irradiated surface ice exhibited a greater
proteinaceous signature than DARK, indicating that bioavailable DOM was not as readily consumed in these
incubations. Bacterial communities in surface ice are genetically diverse (Perini et al., 2019) and as such, are
likely capable of consuming a range of substrates (Fernández-Gómez et al., 2013; Kirchman, 2003). Thus, the
community may be consuming different DOM components across light treatments and it is possible that
genetically distinct communities develop as a result (Mahmoudi et al., 2017; Smith et al., 2018). We also observed
much greater increases in bacterial abundance in irradiated surface ice between 6 and 17 days of incubation
compared to DARK. Solar radiation has been observed to increase nitrogen, sulphur and phosphorus
bioavailability in organic compounds (Antony et al., 2018) and the formation of inorganic nitrogen on exposure
to UV radiation has been widely reported (Bushaw et al., 1996; Wang et al., 2000; Xie et al., 2012). Irradiated



surface ice may have contained a greater nutrient concentration that stimulated more rapid bacterial growth.
Despite this, the quantity of carbon consumed by bacteria did not differ across light treatments indicating that
consumption may be limited by a factor other than DOM composition, such as temperature; however, further
research is required to understand this. Our results therefore indicate that although photodegradation does alter
DOM composition, bacteria in the surface ice are adept at utilising a range of carbon sources to facilitate growth.

**5     Conclusions**
The results from this study reveal the complex interaction between photodegradation and biodegradation in
altering the composition and quantity of secondary phenolic pigment (purpurogallin carboxylic acid-6-*O*-β-D-
glucopyranoside) extracted from glacier algae and surface ice DOM. Both carbon sources are susceptible to
photodegradation, particularly on exposure to UV radiation, which caused the largest compositional changes to
DOM. This is especially important given the potential for ozone holes over the Arctic and subsequent extreme
levels of UV radiation that may result (Manney et al., 2011). Our results indicate that glacier algae secondary
phenolic pigment contains components that can be degraded by surface ice bacterial communities; however,
degradation may be metabolically intensive and therefore pigment is likely not the primary source of carbon
within this system. We also hypothesise that glacier algal pigment may exhibit antimicrobial properties which
inhibit the growth of specific bacterial species. In contrast, surface ice DOM supported extensive bacterial growth
likely due to the wider variety of DOM compounds available. Despite compositional changes to both glacier algal
phenolic pigment and surface ice DOM following photodegradation, we did not observe any difference in
consumption by the bacteria community suggesting that the bioavailability of DOM was not influenced by
exposure to UV or PAR. Photodegradation and biodegradation of surface ice DOM are likely intimately linked
within the surface ice habitat and act as fundamental controls on the composition and quantity of DOM exported
to downstream environments.

*Data availability*
All data is available via the Polar Data Centre (https://www.bas.ac.uk/data/uk-pdc/). DOI TO BE MINTED
AFTER REVIEW PROCESS.

*Team list*
The Black and Bloom team comprises of: Martyn Tranter, Alexandre Anesio, Marian Yallop, Christopher
Williamson, Ewa Poniecka, Miranda Nicholes, Alexandra Holland, Liane Benning, Jim McQuaid, Stefanie Lutz,
Jenine McCutcheon, Andy Hodson, Edward Hanna, Tristram Irvine-Fynn, Joseph Cook, Jonathan Bamber,
Andrew Tedstone, Jason Box and Marek Stibal.

*Author contribution*
MN, CW and AA conceived and designed the study. AH aided MN with sampling during the incubation
experiment. Sample analysis and data presentation was conducted by MN with supervision from CW, AA, MT
and MY. MN wrote the manuscript with inputs from CW and AA, all authors reviewed the final manuscript.




*Competing interests*
The authors declare that they have no conflict of interest.

*Financial support*
This work was funded as part of the UK Natural Environment Research Council Consortium Grant "Black and
Bloom" (NE/M021025/1).

*Acknowledgements*
The authors would like to thank the entire Black and Bloom team, especially those involved in the 2017 field
season. We also thank Jennifer Maddalena, Helena Pryer, Fotis Sgouridis and Ioanna Petropoulou for their
assistance with experimental set up and supporting using laboratory instruments.

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




**6    Appendices**
Appendix A

**Table A1: Bioavailable DOC (BDOC; %) and bacterial growth efficiency (BGE; %) for the pigment and surface ice**
**across light treatments and in the normal and +nutrient treatments (mean ± SE, n=5 for BDOC and n=3 for BGE).**
**Significant differences were only found in %BDOC for pigment across light treatments per nutrient treatment (i.e.**
**normal or +Nutrients) denoted lower case letters and between nutrient treatments per carbon source, denoted by upper**
**case letters.**

| Carbon source | Light | BDOC (%) | | BGE (%) | |
|---|---|---|---|---|---|
| | | Normal | + Nutrients | Normal | + Nutrients |
| Pigment | UV | [a] 21 ± 8.1 [A] | [ab] 27 ± 6.0 [A] | 2.9 ± 3.7 | 7.9 ± 4.1 |
| | PAR | [a] 21 ± 4.2 [A] | [a] 6.4 ± 17 [A] | 1.9 ± 1.4 | 3.1 ± 1.1 |
| | UV+PAR | [b] 0 [A] | [ab] 28 ± 3.0 [B] | 7.8 ± 2.7 | 2.4 ± 1.4 |
| | DARK | [ab] 9.3 ± 6.2 [A] | [b] 39 ± 8.6 [B] | 0.1 ± 1.3 | 3.4 ± 2.5 |
| Surface ice | UV | 27 ± 0.9 | 32 ± 1.8 | 2.3 ± 0.8 | 4.1 ± 1.2 |
| | PAR | 33 ± 1.2 | 30 ± 2.3 | 11 ± 0.6 | 5.8 ± 2.8 |
| | UV+PAR | 31 ± 2.3 | 31 ± 2.0 | 7.0 ± 2.3 | 13 ± 7.2 |
| | DARK | 33 ± 2.5 | 35 ± 3.5 | 6.7 ± 2.5 | 3.5 ± 0.6 |
