# Peer review of "Photodegradation and biodegradation of dissolved organic matter on the surface of the Greenland Ice Sheet 3"

_Biogeosciences, 2020_

## Referee Comment (RC1) · Anonymous Referee #1 · 6 Jul 2020

This paper examines the photodegradation and biodegradation of algal-derived dissolved organic matter (DOM) on the Greenland Ice Sheet (GrIS). Filtered bulk-ice-melt and extracted-pigment samples were exposed to artificial radiation tracking the evolution of DOC, absorbance and fluorescence of DOM. The light-exposed samples were subsequently inoculated with microbial isolates from the GrIS. This is a well written paper. The methods are largely sound, but I have one major concern regarding self-shading (inner filter effect). Basically, given the large absorbance values, I wonder if the CDOM nearest the light was shading the CDOM behind it which means that the absorbance and photodegradation were underestimated. It is my understanding that the filtered ice and pigment extracts were placed into "250 mL pre-combusted Pyrex

crystallising basins" which were placed directly under the lamps [Sections 2.1 to 2.3]. How deep was the sample in these basins (pathlength)? Given absorption coefficients of 200 /m at 254 nm (Figure 1a) and a 1 cm pathlength, the absorption coefficient was likely underestimated by 57 % [Hu et al., 2002, Limnol. Oceanogr., 47(4), 2002, 1261–1267]. The same calculation for the pigment-extracts (a254=50 /m) gives a 21 % underestimation. In these calculations I guessed a pathlength of 1 cm (very likely for the absorbance and fluorescence measurements), but this would have to be <1 mm for my concerns to be invalid. If the absorbance was underestimated, then the fluorescence correction in section 2.5.1 and SUVA values are also in error. These errors may be similar for the subset of ice treatments or the subset of pigment-extracts (similar absorbance within each subset, therefore consistent error), but any comparison between the two subsets is invalid. I therefore read the rest of the results and discussion rather superficially (as they will likely change). This is a good study and I would like to see it published, but I would like the authors to reprocess their data [Hu et al., 2002] as it may alter their results substantially.

---

## Referee Comment (RC2) · Anonymous Referee #1 · 6 Jul 2020

Specific Comments 1) Line 89: spelling - "remain" instead of "remains" 2) Sections 2.1 and 2.2: Was the pH of the ice and pigment-extracts measured? This would be useful information in the context of e.g. Anesio and Graneli (2003) [Limnol. Oceanogr., 48(2), 2003, 735–744]. 3) Section 2.4 and throughout: I prefer to use the term "microbes" rather than "bacteria" in order to describe both bacteria and archaea. 4) Section 2.4: It would be useful to know the time interval between photodegradation and biodegradation experiment and to have some discussion of potential inhibition of microbial activity by long-lived oxidants which are photochemically produced (e.g. H2O2). One of the authors has previously shown that these longer-lived oxidants may act as stressors to microbial communities and even inhibit growth [e.g. Anesio

et al., 2005, doi:10.1128/AEM.71.10.6267–6275.2005]. If the biodegradation experiments didn't start until a few days after the end of photodegradation experiments, then this wasn't an issue. 5) Line 258: The text states "10 % greater absorption at "250nm and between 325 - 600nm." – this is not evident from Figure 1. Is the Figure correct?

---

## Referee Comment (RC3) · Anonymous Referee #2 · 26 Aug 2020

The authors of "Photodegradation and biodegradation of dissolved organic matter on the surface of the Greenland Ice Sheet" present a research project to survey impacts of one mechanism effects on the other when organic matter, microorganisms, and nutrients are available of one location in southwest Greenland. This work highlights carbon transformations after photodegradation and after biodegradation of the pho-todegraded products (photodegradation after 2 days of different light treatments) to discern bioavailabilities of organic matter to microbial communities. Importantly, recent work has suggested that photodegradation may be producing organic matter products that are more susceptible to microbial degradation, and this is where the project design is strong, and the findings will be impactful even though the work is very localized to

a specific spot of southwest Greenland. This work also specifies that different types of surface ice components may be more or less susceptible to each degradation pathway, which is also important to delineate. Beyond those points, it was challenging to evaluate the data presented in its current form because the methods section lacked important information needed to understand the results and figures. Also, without a procedural blank that is exposed to the WhirlPak bags, it was impossible to evaluate any optical property and carbon concentration data since WhirlPak bags can introduce carbon contamination and have distinct absorbance and fluorescence signals. This may be a fatal flaw in the data set and absolutely needs to be addressed. The introduction mostly sets up the project goals smoothly, with clearly defined background and structure for the paper, however, the methods section and results section did not follow smoothly and at times suggested a lack of understanding of the data. If the manuscript is revised to include more appropriate methods information and the results section is restructured to promote a clearer understanding of the findings, the strength of the paper will greatly improve and be more appropriate for publication. Therefore, I focus this review mostly on the methods section as the highest priority to revise first. After that major revision, the remaining text can be more thoroughly reviewed.

In the following sections major comments are provided starting with the methods section and other detailed comments are provided to page 15 of the manuscript.

Major revisions: 1. Methods section: Details regarding where and how the samples were collected were very strong, however the presentation of the experimental design, number of samples, method controls/appropriate controls, and how values were calculated on what datasets were weak. Consider a conceptual image or data flow chart/table(?) to help promote clarity of the experimental design. Some of the sample numbers are provided but do not follow a pattern that makes sense. 2 carbon sources and 3 light treatments is n=5 photodegradation samples? Shouldn't that be 6? Light treatments equate to 2, 10, and 4 days of exposure? These treatments extended for 48 hours? Were the appropriate method controls used? MQ water blank tells you what?

Was a procedural blank incorporated? It seems to suggest the DARK sample is an abiotic dark sample, is that correct? Be specific, otherwise it changes the way the data is interpreted. Were blanks subtracted? Whirlpak bags have distinct absorbance and fluorescence signals. If that is not consistent across labs or clean vs dirty conditions, that's acceptable, but it cannot be confirmed without measuring it. If procedural blanks (including through everything the sample touched, so bottles, filters, syringes) were not taken in the field, ones can be created in lieu of being physically in Greenland and then subtracted from your data set appropriately. They, at least, get close to helping resolve the data into GrIS signals while reducing confounding contamination variables. Linked to that, the surface ice DOC concentrations were amended or actually measured to be ~18ppm? That's a lot of carbon in the surface ice world and needs to be appropriately explained. What information is gained from the MQ water controls? For photodegradation, it makes the most sense to subtract out an abiotic dark control. Was one used or three? One for each scheme? For biodegradation in the dark, following photodegradation, the dark controls were also incorporated? All of biodegradation took place in the dark, correct? Was another control created that is an abiotic dark biodegradation control? Amended in the same way with microbes for both carbon sources, but then filtered to remove other abiotic transformations? How can the authors be sure other abiotic reactions weren't taking place in the biodegradation experiments? More information needs to be provided to explain this or present the results in a way that includes such a caveat when interpreting the information. More details on what values were calculated from what data need to be included to better understand the figures. This can be clarified in the text and then should follow clearly to the figures. For example, how were the peak region boundaries discerned? Table 2? Explicitly state that, since based on Figure 3, there aren't separate boundaries for B/T. Separate boundaries exist and this should be made clear in the methods and presentation of results. Another example is, what data was used for Figure 6? Three data points for each light treatment per carbon source and in both transformation mechanisms, but the methods section does not describe how to make sense of this in photodegradation, biodegradation, and

fluorescence spectroscopy sections.

2. It's important to extend carbon transformation research to include both photodegradation and biodegradation, however, we know these mechanisms do not occur in sequence, they are occurring all the time together. Therefore, this project should target more of why a sequential experimental design was used, what that data helps us understand, and how it can be appropriately interpreted in the context of natural processes occurring on the GrIS. Please provide more context and exercise caution when citing references that do not include appropriate controls when assessing photodegradation and biodegradation pathways.

3. The types of bottles used for photodegradation are very important since borosilicate glass filters out specific wavelengths of sunlight. Were the photodegradation experiments conducted in closed bottles? Shaken? Open bottles? Bulbs were used from directly above the bottles? This is an important aspect to include since bottle lids would cast a shadow. Please provide more context and exercise caution when citing references that use glassware that filters out sunlight and has bottle lids casting a shadow when assessing photodegradation pathways.

4. It appears that the experimental design using a coupled destructing sampling scheme over time with the "before and after" approach leads to more confusing questions than it helps answer about processes. Why were two approaches used instead of one? Using a destructive sampling scheme over time for microbial abundances and carbon concentrations suggest that it was possible to also sample for optical spectroscopy to help characterize the organic matter over time and not just at the end. This seems to be a major component of the paper, understanding organic matter from various processes. If only end products are to be considered, what is the point of measuring carbon concentrations and microbes throughout? How do these things tie together? In its current form, it suggests that portions of this experiment were well developed while others were not. More context will need to be provided to make the argument for "before and after" understanding when temporal sampling and analyses

were available.

5. Not enough details were provided in the fluorescence spectroscopy section and each sentence seems to speak to different themes that require more context/information. Three are apparent: instrumental settings, processing/generating the EEMs, and organic matter characterization. Please include integration times, fluorescence intensity information (normalization, daily checks, or corrections/units), and how many samples were analyzed. In terms of the blank, MQ water is a cuvette, water, and instrument check, not a procedural blank for your work. See the previous sections outlining how important it is to subtract the correct blank, especially when using plastic, whirlpak bags, and filtering samples. Indicating what part of the absorbance scans are used for inner filter effects will be very helpful here and include the appropriate reference. Were any samples diluted prior to EEMs from the absorbance data checks? With carbon concentrations ∼18ppm, they would likely need to be. Please check this. Lastly, organic matter characterization probably should get its own section seeing as two tables are referenced in it, although the use of both of them seem to be in appropriate in the main text and could easily be summarized and moved to the SI. Table 1, in particular, reads very awkwardly and has been generated in a great deal of papers over the last two to three decades. If there is a need to use a reference table, consider extending information from the ones constantly published like this, so that it goes beyond what is already known. This point is even more applicable/important for Table 2, which is a near identical reproduction from Fellman et al., 2010's paper. Were organic matter assessments provided for the starting material? What was it before degradation? What are the concentrations at the beginning of the experiment that go with this information?

6. The overall structure of the data presented in the first few results paragraph would greatly benefit from a reorganization, potentially flowing the way the revised methods section would suggest upon revision. Why are the dark samples presented first? It suggests that the controls are presented first with comparisons to the data from the

experiment. Would it make sense to begin with photodegradation and subheadings of each treatment and what measurement? These paragraphs were hard to follow in their current form and did not accompany what the figures were showing at times. In other instances, some Figure reference on where this result points to for visual references would have been very helpful. Taken together, it was difficult to evaluate whether the data presented were being interpreted correctly and if it made sense.

7. Do the Figures and Tables target what is needed to appropriately represent the results? The Figures will need to be improved to remove background colors, grid-lines, and similar color schemes with the data. They are very difficult to read with gray backgrounds. R default colors are not colorblind not grayscale safe. Figure and Table captions are vague and do not explain enough information presented. Consider improvements to reduce background confounding colors and features, add lettering schemes to point to each panel, more accurate labeling schemes (peak labels), and more context in the captions.

Minor Comments to page 15:

Title: Consider a more specific title to the locale of the study area. This title seems to imply a lot of information of the GrIS.

Text throughout: Consider using the oxford comma to improve clarity. The first example is suggested for Line 13: add a comma after "transformation"

References throughout: There is inconsistent formatting of italics for et al. in the text. Please check for consistency.

Lines 39-40: Are these definitions the best ones relevant to this work? How does this fit in with characterizing organic matter after degradation?

Lines 42-43: Typo? "is usually comprised?"

Line 52: Consider adding a comma after "compounds"

[Figure]

Line 65: Typo? Reduces?

Lines 96-97: Check the Antony references for appropriate dark and blank controls for data interpretation.

Lines 75 and 113: Check the paper for consistency among "south west", "southwest", and "south-western" usages.

Line 114: Please add a space before "cm"

Line 125: Please add a comma after "frozen"

Lines 153 and 156: Please add a comma after "ice"

Line 163: Please add a comma after "17"

Line 167: Please add a comma after "sources"

Lines 166-180: Are the absorbance data used in the calculation available/reported in the SI?

Line 179: Please delete the extra space between "be" and "inferred"

Line 184: This reference is not the original one and seems inappropriate with the lists of original fluorescent and absorbance references provided. Please correct.

Lines 184-186: This sentence is incorrect as written and suggests a lack of understanding of characterization. Fluorescence spectroscopy therefore characterizes fluorescent DOM? It is the information/signals (fluorophores) of fluorescence spectroscopy that can be used to characterize FDOM in terms of potential source, reactivity, and composition? Please correct. A comma is recommended after the word "reactivity".

Lines 186-187: This sentence is incorrect as written. If temperature effects were minimized, then the samples would have been run closer to their natural habitat temperatures ($\sim$3 C?) rather than warming them to room temperature. To minimize the effects of temperature on the already melted samples, they were all run at the same temperature, but as written, this calls into question all the temperature swings these samples have already faced and appropriate statements should address that fact if care was taken to reduce temperature effects.

Line 193: Correct "was identified in sample EEMs" to "was identified from sample EEMs"

Line 203: DOC concentrations were only measured following degradation? This is inconsistent throughout the manuscript. A blank control would be very necessary to incorporate here. Freezing and thawing processes can change the carbon concentration and composition. What was the need for freezing and thawing prior to analysis? This seems like a lot of temperature stress and it needs to be addressed.

Line 212: Procedural blanks? What were they?

Line 216: The start and end of the 31 day biodegradation? This is not consistent with previous text.

Line 220: Please add a comma after "15". Why are these time points different than the biodegradation points?

Line 222: Please add a comma after "filtering"

Lines 254-257: Two peaks? How are small vs. large abs coefficient data assessed? Dark sample information first? Peaks were highly comparable? Conspicuous? What does that mean? Depression?

Line 266: The peak was absent? Meaning zero? Or just no increase in abs? Developed?

Line 272: Where can the 20% increase be seen?

Figure 1 (and all figures using ggplot in R with default backgrounds): Please remove the gridlines and gray background. These figures are very difficult to read with gray on gray colorscales from the background and dashed lines. Please consult the colorblind and

grayscale color suggestions in R to improve readability. These figures show significant differences? Where? These data are averages of what? Are the first derivative data reported anywhere? Discussed? Consider using lettering schemes for both panels in a way where the specific figure panel is pointed to from the text, e.g., Figure 1a-d and Figure 1e-h.

In general, these first paragraphs of the results were very difficult to follow and evaluate.

Section 3.1.2. was also very hard to follow and some language suggested a weak grasp on the understanding of organic matter characterization from fluorescence spectroscopy. Please consider language of amino acid-like DOM instead of proteins/protein-like or proteinaceous without further explanations. Peaks B and T do not fluoresce in the same region, so showing the EEMs in this section with a "B/T" label is incorrect and will mislead readers into interpreting them. They specifically correspond to different types of DOM. Peak M is discussed, but not included in Figure 3? Consider showing the EEMs data before the summary of all the peaks in Figure 2. What is it first, then show how it is being characterized. What data is Figure 2 calculated from?

Lines 317-318: How were dominant peaks determined? Predominant nature? Just presence or absence, or fluorescence intensities?

Lines 337-338: The average HIX value with its SD for surface ice falls within the range of the pigment samples. How are they significantly different? Please check this section.

Figure 4: What is gained from plotting HIX vs FI in these sequential degradation schemes? Do changes in the dark controls for HIX and FI make sense?

―――――――――――――――――